# Redevelopment and Examination of the Psychometric Properties of the Chinese Version of the Last 7-Day Sedentary Behaviour Questionnaire (SIT-Q-7d-Chi) in Hong Kong Older Adults

**DOI:** 10.3390/ijerph19105958

**Published:** 2022-05-13

**Authors:** Ka Man Leung, Ming Yu Claudia Wong

**Affiliations:** 1Department of Health and Physical Education, Education University of Hong Kong, New Territories, Hong Kong, China; leungkaman@eduhk.hk; 2Department of Sport, Physical Education and Health, Hong Kong Baptist University, Kowloon, Hong Kong, China

**Keywords:** sitting, sedentary, inactivity, older adults, Chinese

## Abstract

(1) Background: This study examined the psychometric properties of the Chinese version of the Last 7-Day Sedentary Behaviour Questionnaire (SIT-Q-7d-Chi) in Hong Kong older adults; (2) Methods: Study 1 assessed the questionnaire’s test–retest reliability, and Study 2 examined its validity. Place the question addressed in a broad context and highlight the purpose of the study; (3) Results: In Study 1, 84 older adults (aged 60–90) completed the SIT-Q-7d-Chi twice over a 2-week interval, and in Study 2, 38 older adults (i) completed the SIT-Q-7d-Chi and the Sedentary Behaviour Questionnaire for Older Adults (SBQOA) and (ii) wore a waist-mounted accelerometer for 7 consecutive days. In Study 1, the SIT-Q-7d-Chi’s test–retest reliability (ICC = 0.91–0.99) was satisfactory, and adequate internal consistency was found for most domains of the SIT-Q-7d-Chi (Cronbach’s alpha value being 0.7 or above). Study 2′s results showed that the SIT-Q-7d-Chi results were significantly correlated with the SBQOA results, but not with the accelerometer results; (4) Conclusions: This study revealed the prevalence of sedentary behavior among Hong Kong’s senior citizens, which can be used as a reference to plan or evaluate a future sedentary behavior intervention for older persons, including identifying the content and intensity of activities.

## 1. Introduction

### 1.1. Hong Kong Aging Population and the Phenomenon of Physical Activity

Population aging is a growing concern in Hong Kong. According to the Hong Kong Population Projection Statistics for 2017–2066 [1], the older adult population, which was 1.16 million in 2016, is expected to increase to 1.82 million in 2026, with a massive increase of up to 2.37 million estimated in 2036 and 2.59 million projected by 2066. It is estimated that adults aged 65 years or older will comprise 37% of the overall Hong Kong population within 20 years. The Health Exercise for All Campaign [2] revealed that only 30% of older adults in Hong Kong perform sufficient physical activity (PA); approximately 60% of older adults are overweight or obese. The Behavioural Risk Factor Survey conducted by the Department of Health (2011) [3] also revealed that 21.2% of the surveyed adults aged 55–64 years had sedentary lifestyles. The results showed that older adults were sedentary for approximately 5–8 h or more per day during weekdays. Studies [4,5] have shown that physical inactivity negatively affects older adults’ physical health. The World Health Organization (WHO) (2010) stated that physical inactivity is a leading risk factor for noncommunicable diseases (NCDs) and death worldwide. Physical inactivity may also increase the risks of cancer, heart disease, stroke, and diabetes by 20–30% and shorten one’s lifespan by 3–5 years compared with adequate PA [6].

### 1.2. The Relationship between Sedentary Behavior and Physical Health

The relationship between sedentary behaviour and health has been examined in various studies [7,8,9]. Sedentary behaviour has been positively associated with the risk of cardiovascular disease, and the risk differences among people with diverse sedentary behaviours were up to 80% [7,8]. Studies have also indicated that sedentary behaviours increase the risks of being overweight and developing high abdominal fat [9]. Copeland et al. (2017) [10] conducted a critical review of the consequences of sedentary time on geriatric-relevant health outcomes in older adults. The results showed that more sedentary time in older adults was associated with lower muscular strength, greater limitations in activities of daily living, increased fall risk, cognitive decline, higher risk of adverse mental health outcomes, and lower quality of life. Importantly, empirical evidence supports that both physical inactivity and sedentary behaviours are separate constructs and that these constructs independently predict health outcomes [11,12,13].

Sedentary behaviour refers to activities involving sitting and a low level of energy expenditure that is not substantially above the resting level, in which the energy consumed is ≤1.5 metabolic equivalents (METs*) [14]. Lying down, sitting, watching television, and using a computer are examples of sedentary behaviour. A health promotion survey in Hong Kong revealed that physical inactivity and sedentary behaviour would burden society through the hidden and growing medical care costs as well as through the loss of productivity [15]. In China, sedentary behaviour was found to be associated with an increase in annual out-of-pocket healthcare expenditure on outpatient care, inpatient care, medication, and formal caregiver expenses, with approximately USD 37 incurred for each additional sedentary hour [16]. Therefore, further studies on sedentary behaviour should be warranted. In 2017, Dogra and colleagues [11] developed a consensus statement to integrate perspectives on current knowledge and expert opinions pertaining to sedentary behaviour in older adults into different topics (such as intervention and measurement). The validity and reliability of self-reported measures of sedentary behaviours are crucial research topics.

### 1.3. Sedentary Behavior Measurements

Various sedentary behaviour measurement tools are described in the literature, including subjective (e.g., self-reported measures) and objective (e.g., accelerometers) measures. Accelerometers (e.g., Actigraph) have been increasingly used in research settings to objectively monitor sedentary patterns [17]. Gorman et al. (2014) [17] undertook a systematic review of studies that used Actigraph accelerometers to assess sedentary behaviour in older adults. In the review, five cut-off points were used for classifying sedentary time, and a correspondingly large range of sedentary time was found (62–86% of the day). The majority of studies (48 of 59) adopted the same cut-off point for measuring sedentary behaviour using accelerometers, in which Actigraph accelerometers were set at 1 min epochs (40 of 59) for ≥7 days (52 of 59) [17]. With the various cut-off points for justifying the level of sedentary behaviour, it could be assumed that older adults from different countries have various living patterns. The unstandardized cut-off points may affect the implications among different samples, which would reduce the reliability and validity of the measures in a particular culture. Therefore, despite accelerometers being considered the gold standard of measuring physical activity level, without a standardized cut-off points for measuring sedentary behaviour is considered as a limitation. Hence, the reliability and validity of sedentary behaviour subjective measures should be examined as well.

Subjective measures such as self-reported questionnaires can be used for measuring sedentary behaviour and can be easily adapted among large sample studies. However, self-reported questionnaires may have limitations such as recall bias, emotional influence, literacy deficiencies, or other response-related difficulties [18]. Notwithstanding, questionnaires are considered more cost-effective and less time-consuming. Moreover, subjective sedentary behaviour measures are important for examining the context (what, when, where, and with whom) and type of sedentary behaviour [17,19], which are not typically measured using objective measures. Moreover, subjective measures can be standardised and validated after translation and thus applied across cultures.

### 1.4. The Last 7-Day Sedentary Behaviour Questionnaire (SIT-Q-7d)

Various subjective sedentary behaviour measures are available, such as the Marshall Sitting Questionnaire [20], Measure of Older Adults’ Sedentary Time [21], and LASA Sedentary Behaviour Questionnaire [22]. Medicine National Institutes of Health in America use the Last 7-Day Sedentary Behaviour Questionnaire (SIT-Q-7d) to measure adults’ sedentary behaviour, and this questionnaire is similar to the International Physical Activity Questionnaire; it reports sedentary behaviour in the past 7 days [23]. The SIT-Q-7d is a comprehensive questionnaire that provides both total and domain-specific sedentary behaviour estimates with acceptable reliability and relative validity in non-Asian populations [24]. It is also important to note that sleeping is included as one of the main-specific sedentary behaviour elements in the SIT-Q-7D [23], which other subjective measures, as well as objective measures, do not cover. Despite SIT-Q-7D being sleeping-inclusive, these psychometric properties support the effective use of the questionnaire for population monitoring and in observational studies investigating the overall and domain-specific patterns of a population as well as their associations with health-related outcomes [23,24]. The questionnaire’s reliability and validity have been examined in different populations such as Belgian adults [25], Spanish young adults [24], and Flemish adults [25]. For example, Felez-Nobrega et al. (2019) examined the criterion validity of the SIT-Q-7d in a subsample of the Peninsular Spanish population. The Spanish version of the SIT-Q-7d (SIT-Q-7d-Sp) provided results similar to the those of AP3M +Log for meal, work, and transportation-based sitting time on weekdays, weekend days, and over the previous 7 days.

The SIT-Q-7d has not been applied in the Chinese population, including in Chinese older adults. Bao et al. (2020) [26] identified gaps in sedentary behaviour research in a Chinese context and proposed recommendations and directions for sedentary behaviour research and policy practice in China. One key recommendation was to improve the accuracy of sedentary behaviour measures (including both subjective and objective measures). This was proposed because, among the studies included in the review, only 27.8% reported the validity of the questionnaire(s) used. Additionally, Bao et al. recommend that researchers examine the determinants and outcomes of sedentary behaviours in population subgroups, such as older adults. Sedentary behaviour research related to older adults is in the emerging stage; further studies are warranted to examine the psychometric properties of the SIT-Q-7d in the context of Chinese older adults.

### 1.5. Purpose of Study

In order to address the aforementioned research gaps and Hong Kong’s ageing population, this paper addresses the underdevelopment of sedentary behaviour measurement tools to analyse a Hong Kong population, specifically older adults. In this study, the psychometric properties of the SIT-Q-7d sedentary behaviour measure are examined in a Chinese older adult population.

This is the first Chinese-translated subjective sedentary measure relevant to Hong Kong older adults. It may enable an additional assessment of sedentary behaviour in Hong Kong and Mainland China and could be especially applicable for measuring sedentary behaviour among older adults.

## 2. Materials and Methods

### 2.1. Participants

The target population of the study was adults (1) aged 65 years or above (2) with no diagnosed cognitive impairment and (3) with the ability to read or communicate in Chinese. A convenience sampling method was adopted to recruit older adults from 3 to 4 elderly centres in New Territories, Kowloon, and Hong Kong Island (the three territories of Hong Kong). Those with a score of less than 6 on the Abbreviated Mental Test [27] were excluded from the study. Based on the outcome of a comparison of self-reported and the device measured sedentary behaviour meta-analysis [28], the meta-analytic correlation coefficient, r = 0.32, was taken as the potential power of the reliability and validity examination of the current research. Using the G*Power calculation, it indicated that 74 participants are required for the scale redevelopment and examination.

Study 1: In total, 104 older adults agreed to participate in Study 1. All of them completed the questionnaire, and 84 of them completed the same questionnaire in a retest after a 2-week interval.

Study 2: A total of 40 participants (8 men and 32 women) were included in Study 2, and most participants were within the age range of 75 to 90 years. One participant withdrew from the study, and one outlier was identified. The data of 38 participants (7 men and 31 women) were included in the analysis. The participants’ sociodemographic characteristics (Study 1 and Study 2) are provided in Table 1 (Section 3).

### 2.2. Procedures

#### 2.2.1. Study 1: Reliability Test

Data were collected from participants at elderly centres with approval from the respective person-in-charge (PIC). After we obtained the approval of the centres’ PICs for permission to collect data from their centres, one research assistant explained the purpose and details of the study and answered questions from potential participants before data collection during a site visit. All participants were required to sign written consent forms to declare that they understood the confidentiality agreement of the study and were aware that they could withdraw from the project at any time. After their consent was obtained, they completed the questionnaire in a face-to-face context with the assistance of trained researchers (i.e., research assistants). For a test–retest reliability analysis, participants were required to complete the questionnaire twice; therefore, all participants returned to the centres to recomplete the questionnaire sets after a 2-week interval. Participants spent, on average, 15–20 min completing the questionnaire.

#### 2.2.2. Study 2: Validity Test

The same questionnaire survey data collection procedures of Study 1 were applied in Study 2. However, the participating elderly centres were different to those in Study 1. In the questionnaire, both the redeveloped questionnaire—SIT-Q-7D—and the Sedentary Behaviour Questionnaire for Older Adults were included in examining the concurrent validity. After the completion of the questionnaire, participants who were involved in Study 2 received instructions on how to use the accelerometer and were required to wear the waist-mounted accelerometer for 7 consecutive days, except when bathing, swimming, and sleeping. Through communication with the PICs of the centres, the researcher then collected the accelerometers from the centres after the recording period.

### 2.3. Measures

#### 2.3.1. Subjective Sedentary Behaviour Measure

##### Chinese Version of the Last 7-Day Sedentary Behaviour Questionnaire (SIT-Q-7d-Chi)

The SIT-Q-7d (Wijndaele et al., 2014) [25] is a self-reported questionnaire that measures sedentary time in the last 7 days, including sleeping and napping time. The questionnaire contains 20 items that cover 5 aspects of people’s activities of daily living; they assess sedentary behaviour time (‘sitting or lying down’). The sedentary behaviour includes the time for (a) sleeping and napping; (b) meals, indicating the amount of time spent sitting while having breakfast, lunch, and dinner; (c) transportation (travelling to and from your occupation, travelling as a part of your occupation, and travelling apart from occupation-related travelling); (d) work, study, and volunteering (a summary of the time spent sitting during occupation); and (e) screen time (watching TV, DVDs, and videos, using the computer, and playing computer games) and other activities (reading, household tasks, caring, hobbies, socialising, music, and others). The questionnaire enables the calculation of domain-specific sedentary behaviour time and the total sedentary behaviour time; thus, it emphasises the importance of entering only one specific time for each sedentary behaviour to avoid calculation errors. In the data analysis of concurrent validity, only the total daily sedentary behaviour time was taken into account.

The SIT-Q-7d (English version) was translated, including forward and backward translation, as suggested by Hambleton and Kanjee (1995) [29]. In forward translation, the English questionnaire was translated into Chinese by a translator from a translation centre in Hong Kong. This centre has provided quality translation services for many years. The Chinese version of the SIT-Q-7d (SIT-Q-7d-Chi) was then translated back into English by another translator from the same centre. In both forward and backward translation processes, based on communication with the translators from the aforementioned centre, ambiguous items were identified, and minor discrepancies were identified and resolved through discussion between both translators and investigators. The translators were blinded to the original questionnaire and study purposes. The SIT-Q-7d-Chi was then subjected to content analysis by a panel consisting of five older adults, one statistical expert, and two academic researchers in related areas. After some minor revisions were incorporated, the validation of the final version of the SIT-Q-7d-Chi was conducted.

##### Sedentary Behaviour Questionnaire for Older Adults

The Sedentary Behaviour Questionnaire for Older Adults (SBQOA) [30] consists of 10 items related to sedentary behaviour, such as television watching, using the computer or Internet, reading, travelling, social chatting, engaging in sedentary hobbies, taking a nap, and performing sedentary work-related tasks. Participants were required to record the exact numbers of days and the average time spent on the aforementioned sedentary behaviours instead of providing a range. A standardised formula enables the ‘self-reported daily total sedentary time’ to be quantified by summing all the average time periods spent on each of the 10 items; the formula for estimating the average daily time spent on each item is as follows: (number of days engaged in the behaviour × average time spent in the behaviour a day)/7 [30]. Ku et al. (2016) tested the reliability and validity of the questionnaire in Mandarin among a Taiwanese older adult population and found adequate test–retest reliability; the correlation coefficients of most items ranged from 0.61 to 0.92, and the coefficient for total sedentary time was 0.74. Moreover, accelerometer-derived sedentary time was used to determine concurrent validity, and a significant correlation was found with self-reported sedentary time (r = 0.52).

#### 2.3.2. Objective Sedentary Behaviour Measure

Actigraph wGT3X-BT accelerometers (LLC, Pensacota, FL, USA) were used to objectively monitor participants’ sedentary behaviour. This device provides PA and sedentary behaviour measures, including activity minutes, steps, and PA level proportion through the sensing of movements and vibrations. Participants were instructed to wear the accelerometer over their hip, secured with an elastic belt, during the day for 7 consecutive days, except when bathing, swimming, and sleeping; at least 8 h of recorded activities were defined as a valid day, and data for 3 valid days were extracted for analysis. Moreover, the non-wear time was filtered from the raw data based on a minimum of 90 min of consecutive vector-magnitude counts per minute equal to zero, with tolerance not permitted [31]. According to a recent review of the cut-off points of older adults’ sedentary behaviour by Heesch et al. (2018) [32], the cut-off points in this study were set as follows: sedentary activity, <200 counts/min; light PA, <1951 counts/min; and moderate-to-vigorous PA, >1952 counts/min, with the epoch length being 60 s. These cut-off points were considered to be within an acceptable mean bias range and to enable the precise measurement of older adults’ sedentary behaviour [33]. Finally, a weighted daily mean sedentary time (minutes per day) was computed as follows: ((Σ (sedentary time during typical weekday) × 5) + (Σ (sedentary time during typical weekend day) × 2))/7.

### 2.4. Statistical Analysis

In the current study, all data analyses were conducted using IBM SPSS 25. To determine the reliability of the SIT-Q-7d-Chi in Hong Kong older adults (Study 1), test–retest reliability was measured through the calculation of the intraclass correlation coefficient (ICC). ICCs indicate the repeatability of the responses across the test and retest, and an ICC higher than 0.70 is considered acceptable [34]. Moreover, Cronbach’s alpha coefficient was used to examine the internal consistency of the questionnaire. A Cronbach’s alpha of 0.70 or above is considered adequate. To examine the validity of the SIT-Q-7d-Chi, relationships between the scores of the SIT-Q-7d-Chi and SBQOA [29] and the data collected from the objective measurements (accelerometer) were examined based on an estimation of Pearson’s correlation coefficients. Correlation coefficients of 0.5–0.75 indicate a moderate to close relationship, and coefficients higher than 0.8 indicate a strong relationship [35].

## 3. Results

### 3.1. Study 1: Reliability

Table 1 has showed the demographic information of the participants. Among the 84 older adults, more than 50% were 65–74 years old (63.1%) and women (61.9%); approximately 40% of older adults lived in public housing.

The results indicated that the items in most domains had adequate internal consistency, with Cronbach’s alpha values of 0.7 or above, except for the items for transportation and work. Their Cronbach’s alpha values ranged between 0.5 and 0.6. However, the items of the scale were considered to be reliable.

To determine the test–retest reliability of the SIT-Q-7d-Chi, the intraclass correlation was used to evaluate the responses between the 2-week interval (see Table 2). All the domains had excellent test–retest reliability, with an ICC of 0.9 or above.

### 3.2. Study 2: Validity

Referring to Table 1 as well, among the respondents in Study 2, 31.6% were 75–80 years old, most were women (81.6%), and most had attained primary level education or below (78.9%). Approximately 45% of participants lived in public housing. The normality of all variables used in the data analyses was assessed, and the data were found to be normal.

Table 3, Table 4 and Table 5 present the descriptive statistics of the time spent engaging in different sedentary behaviours based on the SIT-Q-7d-Chi (sleeping time, sitting time during meals, sitting time during transportation, sitting time during work, screen-based leisure sitting time, and sitting time during other activities), accelerometers, and the SBQOA [29].

As shown in Table 3, according to the SIT-Q-7d-Chi, the mean sitting time per day including sleep and excluding sleep was 1086.39 min (SD = 259.76 min) and 573.71 min (SD = 248.68 min), respectively. Next, referring to Table 4 and Table 5, the mean sitting time spent per day, as measured using the SBQOA and accelerometers, was 518.99 min (SD = 249.95 min) and 477.94 min (SD = 160.90 min), respectively. A comparison of the average sitting time spent per day, as measured using the three sources (i.e., the SIT-Q-7d-Chi, SBQOA, and accelerometer), revealed differences of approximately 8 to 19%.

The Pearson’s correlation result was showed in Table 6. It revealed that the total sedentary time measured using the SIT-Q-7d-Chi including and excluding sleep time was positively correlated with that measured using the SBQOA (Po et al., 2016) (r = 0.47, *p* = 0.01; r = 0.43, *p* = 0.01), respectively. Particularly, the screen time based on the SIT-Q-7d-Chi was positively correlated with TV watching time (r = 0.47, *p* = 0.01) and computer use time (r = 0.57, *p* = 0.01) based on the SEQOA. Similarly, the sleep domain of the SIT-Q-7d-Chi was also significantly associated with the nap domain of the SEQOA (r = 0.38, *p* = 0.05). In addition, two domains of the SEQOA (reading [r = 0.34, *p* = 0.05] and chatting [r = 0.37, *p* = 0.05]) that were not included in the SIT-Q-7d-Chi were positively correlated with the “other domain” of the SIT-Q-7d-Chi.

However, the objectively measured results from accelerometers (i.e., minutes per day) had no significant correlation with the SIT-Q-7d-Chi measures, regardless of whether the sleeping time was included or excluded (i.e., min per day, r = −0.15, *p* > 0.05 (including sleeping time); r = −0.07, *p* > 0.05 (excluding sleeping time)). Although the SIT-Q-7d-Chi had no correlation with the objective measure, the significant correlations of the total scores and the scores of some domains of the SIT-Q-7d and SEQOA indicated adequate scale validity. Thus, the SIT-Q-7d-Chi is valid for assessing sedentary behaviour as a whole and in some domains.

## 4. Discussion

This study examined the psychometric properties of the Chinese version of a sedentary behaviour measure, namely, the SIT-Q-7d-Chi, in the Chinese older adult population. Study 1 and Study 2 examined the reliability and validity of the SIT-Q-7d among Chinese older adults, respectively. Our results showed the scale’s internal and test–retest reliability as satisfactory. However, the validation results showed that the SIT-Q-7d-Chi was only significantly correlated with the SBQOA; it was not correlated with the objective measure.

First, in this study, participants’ average sedentary time per day recorded using accelerometers was lower than that in previous studies (See Table 7). This can be explained by cultural differences. Biddle et al. (2019) [36] found that Chinese people are less sedentary than Europeans. Even when we used 200 counts per min as our cut-off point in examining older adults’ sedentary behaviour, our participants’ average sedentary time was lower than that reported in older adults in Western countries. Possible reasons for this include car ownership and cultural norms that affect the sedentary behaviour of Chinese people. Chinese culture emphasises the family unit, and Chinese people embrace the ‘we’ identity. The connection between family members is strong. To reinforce family unity, they organise family meals and celebrations and emphasise the sacredness of genealogy records more frequently [37]. Moreover, car ownership levels in Hong Kong are lower than those in every city in the developed world [38]. All these factors increase older adults’ mobility.

However, SB prevalence in the present study was comparable to that in related studies in Asia. Bao et al. (2020) [26] conducted a systematic scoping review to synthesise sedentary behaviour research related to the Chinese population published from 1999 to 2019. The results revealed that 9.5% of studies reported that in older adults, more than 6 h per day was spent engaging in sedentary behaviour, and the duration of sedentary behaviour mainly ranged from 2 to 6 h a day. Zhang et al. (2014) [39] interviewed 15,193 individuals aged 60 years and over in China and found that Chinese older adults spent, on average, 4.2 h per day engaging in sedentary behaviour. In Singapore, older adults’ total median daily sedentary time was 6.6 h, and 63% of interviewed older adults had less than 8 h of sedentary time [40]. Comparing the illustrated studies to the mentioned research studies, they suggest that, despite the culture of unity, countries such as China or Singapore may be more prone to having high levels of sedentary behaviour due to the necessity of traveling by transport. On the contrary, to our best knowledge, Hong Kong older adults have a tendency to engage in activities or access to daily necessities such as grocery stores, parks, and social gatherings that are within their living neighbourhood or district, within walking distance, or within short transportation periods. Moreover, the Mass Transit Railway is the most popular public transportation in Hong Kong, where mobility during the travel period is still considered high. Furthermore, Hong Kong older adults who engaged in elderly centre and community services tended to be comparably active, and Hong Kong has a higher proportion of hidden older adults who avoid social activities and may not leave the house at all and who may exhibit higher levels of sedentary behaviour.

### 4.1. Reliability

The reliability of the SIT-Q-7d-Chi was satisfactory, and a Cronbach’s alpha value of 0.7 or above was observed for both the test and retest in most domains, with an ICC of 0.9 in all domains. Our results echoed those of previous studies [25]; that is, moderate to high reliability for total scores and moderate to very high reliability for domain-specific sedentary time were found in our study. Notably, the ICC of the work domain was 1. This can be explained as follows: all participants were retired older adults; they did not need to work regularly. Therefore, in line with the recommendations of authors of a validation study of the SIT-Q-7d involving a Spanish population [24], this domain should be excluded in future studies due to its lack of applicability to Hong Kong older adults.

As stated earlier in the text, the internal consistency of the items related to transportation and work was lower than that of other items. Participants were all retired old adults, and their working and transportation schedules fluctuated. Given that most of the retired old adults’ works were learning and doing volunteer job, this working time did not necessarily indicate that they were sedentary despite of the long working hours (i.e., almost 5 days a week). For example, some participants mentioned that they might walk, stand, and sit at different time periods during their work. Therefore, this reduced the internal consistency of the items in these two domains.

### 4.2. Validity

The total sedentary time based on the SIT-Q-7d-Chi may seem more valid than that based on the other domain-specific measures (i.e., the SBQOA). This was indicated by the significant correlation of the total sedentary time and the sedentary behaviour time of some domains of the SIT-Q-7d with those of the SBQOA [30] in our current study. A previous study [24] had partially contrasting results: the SIT-Q-7d (Spanish version) was acceptable for estimating the sitting time in three specific domains only: transportation, occupation, and meal-based sitting time. The aforementioned study also indicated that the SIT-Q-7d-Sp was not appropriate for measuring sedentary time based on total sitting time. However, the results of the SIT-Q-7d-Chi were comparable to those related to total sedentary behaviour time and sedentary behaviour time in some domains of the SEQOA. Possible reasons are the similarity of the question content of the SIT-Q-7d-Chi and SEQOA. First, some domains in these two questionnaires are similar, such as sleep time and screen time. Second, both of them are subjective sedentary behaviour measures. Therefore, significant correlations may arise between the results of the two questionnaires.

In addition, in the validation study, the SIT-Q-7d-Chi did not have significant correlations with the sedentary behaviour time measured using accelerometers. Although accelerometers are typically used to measure body posture, mobility, and gait [47], research has shown that accelerometers are more effective at recording energy expenditure from dynamic activities than for recording body positions such as sitting and standing [48]. Thus, the SIT-Q-7d results may not correspond with actual daily routines recorded using objective measures.

Other validation studies on sedentary behaviour questionnaires have used Bland–Altman plots to indicate the level of agreement between the scale- and accelerometer-estimated times; however, our study showed significant mean differences between the two measures, which violated the inherent rule of thumb of Bland–Altman plots. The results from the scale were not in line with those from the accelerometer.

The SIT-Q-7d has been used to analyse the activities of young adults [25] in general, but not Chinese adults or Chinese older adults. To fill a research gap, this study examined the first Chinese-translated sedentary behaviour measure of sedentary behaviour in Hong Kong older adults. This study also provided information on the prevalence of sedentary behaviour among Hong Kong older adults. Furthermore, the SIT-Q-7d-Chi was validated; it can be applied for investigating the sedentary lifestyle of older adults in Hong Kong in the future. Sedentary behaviour outcomes generated using the SIT-Q-7d-Chi can be used as a reference for developing or evaluating a sedentary behaviour intervention for older adults in the future, including determining the content and the intensity of activities.

### 4.3. Strengths and Limitations

Importantly, we are the first to translate and validate the SIT-Q-7d for older adults in Hong Kong and compare it with another sedentary behaviour questionnaire (i.e., the SBQOA). The findings provide insight into the sedentary behaviour of an Asian population. However, this study has some limitations. First, the unequal gender distribution of the sample in both studies could have limited the representativeness of the sample, but Hong Kong older adults have a slightly higher proportion of women. Second, the majority (approximately 78%) of the participants in Study 2 had only a primary education level or below, which might have affected overall sedentary behaviour, as it has been revealed that sedentary behaviours are more prevalent among individuals with a lower socioeconomic status (e.g., education) [49]. Third, the lack of a domain-specific activity log is another limitation of this study. Finally, in a previous SIT-Q-7d validation study [50], the activePAL3 monitor was found to better distinguish static and dynamic acceleration. Therefore, because older adults have more static movement, the activePAL3 (sp.) monitor can be used for such measurements in the future. In addition, the use of a logbook would enable an in-depth understanding of participants’ sedentary behaviour in different domains. A domain-specific activity log should be used to obtain an in-depth understanding of the sedentary behaviour of participants. In conclusion, this study confirmed the validity and reliability of the SIT-Q-7d-Chi for assessing older adults’ sedentary behaviours. This measure is an additional option for measuring sedentary behaviours among Chinese older adults.

### 4.4. Implications for Future Practice

The SIT-Q-7d-Chi is a specific and reliable sedentary behaviour scale that measures Chinese older adults’ sedentary behaviour in different domains.The field of nursing older adults is given an alternative viewpoint to investigate and examine older adults’ daily activities and habits, thus enabling better day-care and health services for older adultsThe redevelopment of the scale also provides Chinese older adults a self-awareness of the high level of sedentary behaviour, as well as the importance of physical activity during daily practice.

## 5. Conclusions

Through the examination of psychometric properties, this study investigated at the first Chinese-translated sedentary behaviour measure in Hong Kong’s senior citizens. This study also revealed the prevalence of sedentary behaviour among Hong Kong’s senior citizens, which can be used as a reference to plan or evaluate a future sedentary behaviour intervention for older persons, including identifying the content and intensity of activities.

## Figures and Tables

**Table 1 ijerph-19-05958-t001:** The demographic characteristics of Study 1 and Study 2.

	Study 1 (N = 84)	Study 2 (N = 38)
	%	%
**Age**		
60–64	0	2.6
65–69	39.3	15.8
70–74	23.8	7.9
75–80	21.4	31.6
81–84	6	18.4
85–90	9.5	18.4
90 or above	0	2.6
**Gender**		
Male	38.1	18.4
Female	61.9	81.6
Education Level		
Primary or below	-	78.9
Secondary school	-	18.4
Tertiary education or above	-	0
**Housing type**		
Public Housing	40.5	44.7
Home Ownership Scheme	26.2	5.3
Private Housing	16.7	42.1
Others	16.7	5.3

**Table 2 ijerph-19-05958-t002:** Reliability—Internal Consistency and Intraclass Correlation of SIT-Q-7d-Chi (N = 84).

Cronbach’s Alpha Value		ICC	95%CI
Time 1	Time 2		
0.822	0.812	0.986	0.979–0.991
0.866	0.851	0.987	0.981–0.992
0.584	0.593	0.914	0.870–0.943
0.604	0.619	1	1.0–1.0
0.757	0.761	0.984	0.975–0.990
0.698	0.730	0.997	0.995–0.998

**Table 3 ijerph-19-05958-t003:** Descriptive statistics of the average minutes of sedentary behaviour from the Accelerometer.

N = 38	Mean ± Std. Deviation
Total Sedentary Behaviour (min·d^−7^)	3345.58 ± 1126.32
Average 7 Days (min·d^−1^)	477.94 ± 160.90

**Table 4 ijerph-19-05958-t004:** Descriptive statistics of the average minutes spent in different sedentary domains from the Sedentary Behaviour Questionnaire for Older Adults.

N = 38	Mean ± Std. Deviation
Sitting Time during watching TV	163.91 ± 110.73
Sitting Time during using computer	49.87 ± 74.35
Sitting Time during reading	22.21 ± 41.46
Sitting Time during chatting	21.13 ± 39.74
Sitting Time during transportation	17.35 ± 41.73
Sitting Time during meal	107.70 ± 52.20
Sitting Time during hobbies	32.54 ± 60.91
Sitting Time during nap	49.61 ± 96.72
Sitting Time during work or volunteer	28.87 ± 62.44
Sitting Time during other activities	4.17 ± 9.79
Total Sitting Time	518.99 ± 249.95

**Table 5 ijerph-19-05958-t005:** Descriptive statistics of the average minutes spent in different sedentary domains from the SIT-Q-7d-CHI.

N = 38	Mean ± Std. Deviation
Sleeping time	
Weekday (min·d^−1^)	514.08 ± 77.97
Weekend Day (min·d^−1^)	509.13 ± 77.81
Average 7 Day (min·d^−1^)	512.67 ± 77.64
Sitting Time during Meals	
Weekday (mins·d^−1^)	87.38 ± 38.59
Weekend Day (min·d^−1^)	86.97 ± 37.88
Average 7 Day (min·d^−1^)	87.27 ± 38.29
Sitting Time during transportation (*7) (min·d^−1^)	287.83 ± 399.68
Sitting Time during transportation (*day) (min·d^−1^)	65.78 ± 172.97
Sitting Time during work (*7) (min·d^−1^)	356.17 ± 529.85
Sitting Time during work (*day) (min·d^−1^)	224.72 ± 465.70
Screen-based Leisure Sitting Time	
Weekday (min·d^−1^)	227.62 ± 117.73
Weekend Day (min·d^−1^)	223.67 ± 117.32
Average 7 Day (min·d^−1^)	226.49 ± 116.66
Other Sitting Time	
Weekday (min·d^−1^)	152.32 ± 148.51
Weekend Day (min·d^−1^)	173.57 ± 165.99
Average 7 Day (min·d^−1^)	158.39 ± 149.59

**Table 6 ijerph-19-05958-t006:** Validity—Pearson correlation.

Subscale	A	B	C	D	E	F	G	H
A	1							
B	0.121	1						
C	0.153	0.944 **	1					
D	0.166	0.249	0.291	1				
E	0.210	0.476 **	0.501 **	0.065	1			
F	0.411 *	0.454 **	0.419 **	0.537 **	0.581 **	1		
G	−0.003	0.695 **	0.693 **	0.467 **	0.573 **	0.483 **	1	
H	−0.066	−0.148	−0.063	−0.141	−0.180	−0.271	−0.144	1

*. Correlation is significant at the 0.05 level (2-tailed). **. Correlation is significant at the 0.01 level (2-tailed). Note: A: SIT_7d_work_total = Last 7 days Sedentary Behaviour questionnaire total sedentary time during work. B: SIT_7d_total_with sleep = Last 7 days Sedentary Behaviour questionnaire total sedentary time included sleep. C: SIT_7d_total_without sleep = Last 7 days Sedentary Behaviour questionnaire total sedentary time excluded sleep. D: Elderly_SBQ_TV = Sedentary Behaviour Questionnaire for Older Adults_TV time. E: Elderly_SBQ_computer = Sedentary Behaviour Questionnaire for Older Adults_Computer time. F: Elderly_SBQ_total = Sedentary Behaviour Questionnaire for Older Adults_total Sedentary Behaviour time. G: SIT_7d_screen_total = Last 7 days Sedentary Behaviour questionnaire total sedentary time during screen usage. H: Actigraphic_daily_min = Sedentary time per day measured by Actigraphic accelerometer.

**Table 7 ijerph-19-05958-t007:** Objectively Measured Sedentary Behaviour by Accelerometer.

Author	Average Sedentary Time (min/per day)	Cut-off Point
Colley et al., 2011 [41]	600	<100 counts/min
Arnardottir et al., 2013 [42]	624	<100 counts/min
Van der Berg et al., 2014 [43]	618	<100 counts/min
Matthews et al., 2008 [44]	528	<100 counts/min
Clark et al., 2011 [21]	552	<100 counts/min
Bankoski et al., 2012 [45]	570	<100 counts/min
Koster et al., 2012 [46]	540	<100 counts/min
Current paper	477.9	<200 counts/min

## Data Availability

Not applicable.

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
