# Peer review of "Redevelopment and Examination of the Psychometric Properties of the Chinese Version of the Last 7-Day Sedentary Behaviour Questionnaire (SIT-Q-7d-Chi) in Hong Kong Older Adults"

_ijerph, 2022, doi:10.3390/ijerph19105958_

Round 1
Reviewer 1 Report
This article reported on the redevelopment then the reliability and validity of a Chinese version of the SIT-Q questionnaire for measuring sedentary time. The testing was done in Chinese older adults. Overall, the paper would benefit from considerable editing, both the tighten the argument and remove non-directly relevant information, and also to address the number of comments highlighted below.
- Title: suggest using “Redevelopment and Examination” in the title.
The abstract needs careful editing and proof reading:
- Line 12: seems like there is an additional sentence in here remaining from the template “Place the question addressed in a broad context and highlight the purpose of the study”
- Line 15: clarify where the accelerometer was worn, and whether diary data was collected. Also include the age of the participants.
- Line 17 – the word “conclusions” is here, but the results for study 2 are not presented yet. Report the actual validity results for the accelerometer data with the Sit-Q.
- The introduction is currently quite broad, and could be substantially shortened to focus primarily on the evidence gaps in the existing measurement tools, and also why the SIT-Q-7D was chosen. It also needs careful editing, with some of the pick-ups below:
- Lines 23-28 the instruction for the introduction have not been removed.
- Line 82: missing the term adults (…older adults).
- Line 88: clarify what is meant by “particularly in studies with a large sample size”.
- Line 89-91: this sentence is contradictory with both limitations and strengths in the same sentence. Please correct.
- Line 129: Study one examined reliability, study 2 examined validity; however, this line says it is the opposite. Please correct.
- Introduction: The authors argue in the introduction that different lifestyle patterns may need different cut points for sedentary time. However, it is important to note that there are recommendations to move away from cutpoints, and utilise (for example) machine learning methodologies. It is also somewhat unclear how this paragraph relates to the study report.
- Introduction: please explicitly state what the validity of the questionnaire will be measured against in the final paragraph.
- Methods: Lines 145-159 and Table 1 belong in the results as they are reporting on participant characteristics. Please also use formatting in Table 1 to different the headers from the category (e.g., Gender, Men, Women).
- Methods: clarify here whether participants of study were the same as study 1 and if not whether the same recruitment process was used.
- Methods, Line 171: seems like an extra line here (they assess sedentary time)
- Methods: please include sample size calculations.
- Methods: Section 2.2.1.2 – it is unclear at this stage why this questionnaire is listed here, as it was not included in the aims of the study. It is also unclear why this was used as a validity measure, and it is unclear when this questionnaire was asked e.g., was it asked after the sit-Q? The order of the questionnaires could impact on the responses. The methods also imply that only one questionnaire was asked. If there was two, then this should be made explicit in the process section.
- Methods: section 2.2.3 – make clear whether all participants wore accelerometers or just those in study 2.
- Methods: please provide reference and data for the validity of hip worn accelerometers for measuring sedentary time, particularly given these are not the gold standard for field-based measurement. Also please provide reference for the non-wear criterion. Please indicate whether diary data was collected to understand any context-specific information.
- Methods: please provide more detail on the elderly centres and who goes there. Are they representative of the broader older adult Chinese population, including the type and time of sedentary activities that would do?
- Methods: it is unclear why the questionnaire was not asked about the same time period as the monitor wear. This presumes that the behaviour is stable.
- Methods: clarify if the accelerometer data was compared to just the totals, or whether it look at correlations with domain-specific measures. Clarify how wear time was taken into account when analysing and reporting the monitor data.
- Results, line 276-279: this belongs in the methods section.
- Results, line 279; line 285 the descriptions of Table 2 does not match what is presented in Table 2. The Cronbach’s alpha values and ICCs are not presented. This looks more like Table 5
- Results, Table 2: it is unclear why this is n=38 – isn’t this from study 1?
- Results: as per previous comments, it is unclear why the results for the SBQOA are presented as this was not one of the aims of the study. It is recommended to remove this information and association data or to clarify the role more explicitly in the introduction and methods.
- Results: Table 3 shows that accelerometer-derived sedentary time was 3345.58 ± 1126.32 minutes per day. This is not possible – please clarify. Please also add wear time.
- Results, Line 325: as per previous comments, it is unclear why the SBQOA is included, and also then, why it is appropriate to use as a validity measure. These findings do not necessarily indicate true behaviour.
- Discussion, Line 333: recommend not using the word “controversial”. Just report that there was no significant correlation.
- Discussion: there is comparison to previous studies in terms of sedentary time; however, it is unclear how wear time was taken into account. Recommend comparing percentage of the day sedentary across the different studies.
- Discussion: The questionnaires were interviewer-administered. Are there plans to examine these psychometric properties using self-completion?
- Discussion, Line 458-9: this is additional information that should be removed.
- Can the redeveloped version of the questionnaire be added as a supplemental file for use? How are people able to access it?
Author Response
We would like to thank the reviewer for their deliberate reviews of the research article. The raised considerable concerns are very helpful for improving the article. We agree with almost all their comments that we have revised the article and responded to the comments accordingly.
The detailed responses to each of the reviewers’ comments will be stated below. We clearly stated the revised parts with the particular paragraph shown, as well as indicating the page for referring to the paper; or if we have slightly countered with some of the points, we stated the reason with supporting literature. We hope that the reviewers will find our responses persuasive and cogent, and we are willing to accept further suggestions that the reviewers may have.
In the following, each response is targeting each reviewers’ comments, the comments are in italics with responses inserted after it.
This article reported on the redevelopment then the reliability and validity of a Chinese version of the SIT-Q questionnaire for measuring sedentary time. The testing was done in Chinese older adults. Overall, the paper would benefit from considerable editing, both the tighten the argument and remove non-directly relevant information, and also to address the number of comments highlighted below.
- Title: suggest using “Redevelopment and Examination” in the title.
Response: Thank you for your suggestions, changes are done accordingly.
The abstract needs careful editing and proof reading:
- Line 12: seems like there is an additional sentence in here remaining from the template “Place the question addressed in a broad context and highlight the purpose of the study”
- Line 15: clarify where the accelerometer was worn, and whether diary data was collected. Also include the age of the participants.
- Line 17 – the word “conclusions” is here, but the results for study 2 are not presented yet. Report the actual validity results for the accelerometer data with the Sit-Q.
Response: In regard to the suggestions of the reviewer, the required information is added, and the conclusion part is re-written.
- The introduction is currently quite broad, and could be substantially shortened to focus primarily on the evidence gaps in the existing measurement tools, and also why the SIT-Q-7D was chosen. It also needs careful editing, with some of the pick-ups below:
- Lines 23-28 the instruction for the introduction have not been removed.
- Line 82: missing the term adults (…older adults).
- Line 88: clarify what is meant by “particularly in studies with a large sample size”.
- Line 89-91: this sentence is contradictory with both limitations and strengths in the same sentence. Please correct.
- Line 129: Study one examined reliability, study 2 examined validity; however, this line says it is the opposite. Please correct.
Response: Referring to comment 8, It was referred to the adaptability among large sample studies. In regards to the suggestions of the reviewer, the required information is corrected. Please see the highlighted lines.
- Introduction: The authors argue in the introduction that different lifestyle patterns may need different cut points for sedentary time. However, it is important to note that there are recommendations to move away from cutpoints, and utilise (for example) machine learning methodologies. It is also somewhat unclear how this paragraph relates to the study report.
Response: Sorry for the confusion, in order to make our point clear, elaboration has been made upon that. The statement regarding the variations of the cut-off points has been bought up because the unstandardized cut-off points for measuring sedentary behavior might have affected the gold-standard reputation of the accelerometer, and therefore, subjective measures’ benefits should be uphold.
- Introduction: please explicitly state what the validity of the questionnaire will be measured against in the final paragraph.
Response: Thank you for the suggestion. The concurrent validity of the questionnaire is measured, and this is stated at the end of the paragraph.
- Methods: Lines 145-159 and Table 1 belong in the results as they are reporting on participant characteristics. Please also use formatting in Table 1 to different the headers from the category (e.g., Gender, Men, Women).
Response: The demographic information was moved to the result session, and the table is reformatted.
- Methods: clarify here whether participants of study were the same as study 1 and if not whether the same recruitment process was used.
Response: The same recruitment process was use in study 2, as stated in line 246.
- Methods, Line 171: seems like an extra line here (they assess sedentary time)
Response: In regards to the suggestions of the reviewer, the required information is corrected.
- Methods: please include sample size calculations.
Response: Taking a meta-analysis outcome (Prince et al., 2020), the correlation between a self-report and device measure of sedentary time as r=0.32, the Gpower software indicated that 74 samples will be required for this reliability and validity analysis.
Prince, S.A., Cardilli, L., Reed, J.L. et al. A comparison of self-reported and device measured sedentary behaviour in adults: a systematic review and meta-analysis. Int J Behav Nutr Phys Act 17, 31 (2020). https://doi.org/10.1186/s12966-020-00938-3
- Methods: Section 2.2.1.2 – it is unclear at this stage why this questionnaire is listed here, as it was not included in the aims of the study. It is also unclear why this was used as a validity measure, and it is unclear when this questionnaire was asked e.g., was it asked after the sit-Q? The order of the questionnaires could impact on the responses. The methods also imply that only one questionnaire was asked. If there was two, then this should be made explicit in the process section.
Response: Regarding the reviewer’s concern, we have changed the sequence of the methodology, by stating the Procedure before the illustration of the included measure. It is important to point out that another questionnaire is involved because, other than the accelerometer, another well-developed Chinese subjective measure should also be involved in the concurrent validity examination as well.
- Methods: section 2.2.3 – make clear whether all participants wore accelerometers or just those in study 2.
Response: This has been made clear in the manuscript, with only participants who were involved in Study 2 wore the accelerometers.
- Methods: please provide reference and data for the validity of hip worn accelerometers for measuring sedentary time, particularly given these are not the gold standard for field-based measurement. Also please provide reference for the non-wear criterion. Please indicate whether diary data was collected to understand any context-specific information.
Response: Based on the reference below (is indicated in the manuscript), the valid wear days are 3 weekdays and 1 weekend, while the non-wear criterion is having a window of 90-min of consecutive vector-magnitude counts per minute. The current study has minimized the 1 weekend due to the older adults’ participants are retired, the differences between workdays and non-workdays might not be significant.
Choi, L., Ward, S. C., Schnelle, J. F., & Buchowski, M. S. (2012). Assessment of wear/nonwear time classification algorithms for triaxial accelerometer. Medicine and science in sports and exercise, 44(10), 2009–2016. https://doi.org/10.1249/MSS.0b013e318258cb36
- Methods: please provide more detail on the elderly centres and who goes there. Are they representative of the broader older adult Chinese population, including the type and time of sedentary activities that would do?
Thank you for your questions. We understand the reviewer’s concern about the representativeness of the sample. It is important to point out that Hong Kong elderly centers are considered as the most effective and representative way of reaching elderly samples in Hong Kong. All older adults could go to these elderly centers. In Hong Kong, they are considered as a kind of community centers, for those who would like to meet friends, join some free or low-cost leisure activities. These centers are mostly funded by the Hong Kong government and NGOs. Despite reaching older adult participants through the elderly centers are considered as more systematic, in the authors’ opinion, the elderly centers might have a possible effect on the sedentary behavior measure of the older adults. It is because, we are not able to access the hidden older adults, who tend not to access any public activities. While this issue has been added in the discussion section.
- Methods: it is unclear why the questionnaire was not asked about the same time period as the monitor wear. This presumes that the behaviour is stable.
Response: We acknowledge the concern of the reviewer. However, from the practical perspective, not every elderly participant was willing to put on the accelerometer for such as a long period of time. Hence, it caused a certain extent of difficulty for us, and so we need to recruit another group of older adults who were willing to put on the device. But still, the participants of Study 2 have also completed the questionnaire, at the same time period the monitor wear.
- Methods: clarify if the accelerometer data was compared to just the totals, or whether it look at correlations with domain-specific measures. Clarify how wear time was taken into account when analysing and reporting the monitor data.
Response: According to the measurement section in the manuscript, it stated that weighted daily mean sedentary time, while a statement is added to the illustration of the SIT-Q-7D that “In the data analysis of concurrent validity, only the total daily sedentary behaviour time was taken into account.”
- Results, line 276-279: this belongs in the methods section.
Response: The lines were moved to the methods section.
- Results, line 279; line 285 the descriptions of Table 2 does not match what is presented in Table 2. The Cronbach’s alpha values and ICCs are not presented. This looks more like Table 5
- Results, Table 2: it is unclear why this is n=38 – isn’t this from study 1?
- Results: as per previous comments, it is unclear why the results for the SBQOA are presented as this was not one of the aims of the study. It is recommended to remove this information and association data or to clarify the role more explicitly in the introduction and methods.
Response: Sorry for the confusion, the table sequence has been re-arranged by changing originally Table 5 into Table 2, with Table 2 specifying the N=84.
- Results: Table 3 shows that accelerometer-derived sedentary time was 3345.58 ± 1126.32 minutes per day. This is not possible – please clarify. Please also add wear time.
Response: The total sedentary behavior time was the total time of the consecutive 7 days. Sorry for the confusion.
- Results, Line 325: as per previous comments, it is unclear why the SBQOA is included, and also then, why it is appropriate to use as a validity measure. These findings do not necessarily indicate true behaviour.
Response: As mentioned earlier, other than the accelerometer, another well-developed Chinese measure was added to study 2, as an indicator of concurrent validity examination. While it might not be a true reflection of behavior, the SBQOA is a well-developed self-report sedentary behavior measure, which could be compared to the SIT-Q-7D, a newly developed Chinese questionnaire.
- Discussion, Line 333: recommend not using the word “controversial”. Just report that there was no significant correlation.
Response: Thank you for the suggestions. The word controversial is deleted.
- Discussion: there is comparison to previous studies in terms of sedentary time; however, it is unclear how wear time was taken into account. Recommend comparing percentage of the day sedentary across the different studies.
Response: We acknowledge the concern of the reviewer. While, sorry for the confusion that, the comparison unit stated in the discussion (See Table 7) should be clearly stated, which is the daily average sedentary behavior (in minutes), which means the number of days wearing the devices had been divided.
- Discussion: The questionnaires were interviewer-administered. Are there plans to examine these psychometric properties using self-completion?
Response: Sorry for the confusion, to clarify the issue of interviewer-administrated, in fact, the meaning of interviewer-administrated here was having the research assistants read the item word by word to participants of those who are less educated and less literate. Hence, in order to release this confusion, the procedure part has been edited, and restated to “they completed the questionnaire in a face-to-face context with the assistance of trained researchers (i.e., research assistants).” Therefore, there were no individual differences that might have been caused by different interviewers, and the research assistants were just reading the words according to the item, hence, it was still considered as a self-completion questionnaire.
- Discussion, Line 458-9: this is additional information that should be removed.
Response: Thank you for the suggestions. The lines are deleted.
Reviewer 2 Report
This is a high quality paper that is easy to read and understand. The study appears to fill the gap in literature of testing the reliability and validity of the questionnaire on a Chinese population. The introduction provides a good rationale for the study and the methods and results are easy to understand and interpret.
I recommend minor corrections:
Introduction:
Delete lines 23- ‘research’ on 29. This is not relevant to the study so the paper can begin with ‘Population’ on line 29.
Sleep is normally not included in sedentary behaviour definitions so it needs to be clear in the introduction narrative that that is the case.
The introduction could perhaps be tightened and more cohesive - especially the first few paragraphs.
Method:
Participants: Are the participants representative of the Chinese population in terms of demographics e.g. gender? How representative is Hong Kong of the Chinese population?
Line 188- The translators were blinded to the study aims but the explanation for this needs to be developed. Why was this done? Would having context not improve their ability to translate it effectively?
Reliability-
The explanation for the use of interviews to administer the questionnaires needs to be developed further. How do you know the interviewer hasn’t explained the meanings, rather than reading word for word? How was the interview practise standardised to ensure that it was testing the reliability of the measure?
Validity
256- The participants don’t wear the accelerometers whilst sleeping but sleep is assessed in the questionnaire. If the sleep domain is being removed from analysis, like in the results section, this needs to be stated prior in the method section.
Results:
It is mentioned lines 286-287 that the ICC of the domain of work was 1.0 then it is interpreted this is due to participants being retired. This interpretation is mentioned later in the discussion and developed further on lines 367-369. Therefore it can be deleted from the results section.
Discussion:
343-345- This explanation of cultural differences needs to be more explicit. What about the differences mentioned is linked to less sedentary time?
431- ActivPAL3 (sp)
Limitations:
Most of the participants being women (especially in Study 2) should be mentioned as a limitation of the study. (Unless this is representative of the population).
Author Response
We would like to thank the reviewer for their deliberate reviews of the research article. The raised considerable concerns are very helpful for improving the article. We agree with almost all their comments that we have revised the article and responded to the comments accordingly.
The detailed responses to each of the reviewers’ comments will be stated below. We clearly stated the revised parts with the particular paragraph shown, as well as indicating the page for referring to the paper; or if we have slightly countered with some of the points, we stated the reason with supporting literature. We hope that the reviewers will find our responses persuasive and cogent, and we are willing to accept further suggestions that the reviewers may have.
In the following, each response is targeting each reviewers’ comments, the comments are in italics with responses inserted after it.
This is a high quality paper that is easy to read and understand. The study appears to fill the gap in literature of testing the reliability and validity of the questionnaire on a Chinese population. The introduction provides a good rationale for the study and the methods and results are easy to understand and interpret.
I recommend minor corrections:
Introduction:
Delete lines 23- ‘research’ on 29. This is not relevant to the study so the paper can begin with ‘Population’ on line 29.
Response: After reviewing the lines, the lines are deleted.
Sleep is normally not included in sedentary behaviour definitions so it needs to be clear in the introduction narrative that that is the case.
Response: Thank you for your reminder, this narration is added in Line 97-99.
The introduction could perhaps be tightened and more cohesive - especially the first few paragraphs.
Response: After reviewing the lines, the lines are from the journal template which is now deleted.
Method:
Participants: Are the participants representative of the Chinese population in terms of demographics e.g. gender? How representative is Hong Kong of the Chinese population?
Response: Thank you for your questions. We understand the reviewer’s concern about the representativeness of the sample. Firstly, we acknowledge the limited sample size of our research, however, our sample could be considered as somehow representative in terms of geographical coverage by being able to cover all the three territories in Hong Kong. Secondly, it is important to point out that the older adults’ gender distribution in Hong Kong elderly centers is unequal. In Hong Kong, most of the socially active and outgoing older adults are females, hence there are more females recruited from the elderly centers in Hong Kong. Moreover, the gender proportion of Hong Kong older adults was having females more than males. Yet, admitted that this is considered as a limitation of the representativeness of the sample. Still, Hong Kong elderly centers are considered as the most effective and representative way in reaching elderly samples in Hong Kong.
Line 188- The translators were blinded to the study aims but the explanation for this needs to be developed. Why was this done? Would having context not improve their ability to translate it effectively?
Response: Thank you for your question. To our best knowledge, the blinding of research purpose from the translator could reduce the bias of translation, avoid him or her from using wordings that are leaning toward the purpose of healthy lifestyle measurement, and make sure the wordings are lament and can be easily adapted for daily usage.
Reliability-
The explanation for the use of interviews to administer the questionnaires needs to be developed further. How do you know the interviewer hasn’t explained the meanings, rather than reading word for word? How was the interview practise standardised to ensure that it was testing the reliability of the measure?
Response: Sorry for the confusion in using the word “interview”. The use of words in the line of explaining the data collection process is edited, please see line 239.
In fact, the research assistants were only taking an assistant role to read the questionnaire items word by word, instead of having an interview. Also, the research assistants have only assisted the participants when they requested, especially only when there were older adults who were less literate. Therefore, reviewer’s concern on the standardization is minimized.
Validity
256- The participants don’t wear the accelerometers whilst sleeping but sleep is assessed in the questionnaire. If the sleep domain is being removed from analysis, like in the results section, this needs to be stated prior in the method section.
Response: We would like to clarify that the analysis of the accelerometers’ data for the current study has also taken the sleeping time into account. We have assume the period of the non-wearing time during the night and mid-night as sleeping time, and involved in the analysis.
Results:
It is mentioned lines 286-287 that the ICC of the domain of work was 1.0 then it is interpreted this is due to participants being retired. This interpretation is mentioned later in the discussion and developed further on lines 367-369. Therefore it can be deleted from the results section.
Response: Thank you for your suggestions. The respective sentence is deleted.
Discussion:
343-345- This explanation of cultural differences needs to be more explicit. What about the differences mentioned is linked to less sedentary time?
Response: The possible explanation and discussion are included in the discussion session, stating the differences between Hong Kong and other Asian countries.
431- ActivPAL3 (sp)
Limitations:
Most of the participants being women (especially in Study 2) should be mentioned as a limitation of the study. (Unless this is representative of the population).
Response: Despite the Hong Kong government statistics showing Hong Kong females older adults are a slightly higher proportion, the limitation of the gender unequal distribution in the sample is stated in the limitation session in the manuscript.
Round 2
Reviewer 1 Report
The edited version, which uses highlighting and red and green font rather than track changes, makes it very difficult to see the changes made in the document and understand what is the current version. For example, the following text is still included in the start of the introduction, despite specifically highlighting the need to remove this, and other instructions “The introduction should briefly place the study in a broad context and highlight why it is important. It should define the purpose of the work and its significance. The current state of the research field should be carefully reviewed and key publications cited. Please highlight controversial and diverging hypotheses when necessary. Finally, briefly mention the main aim of the work and highlight the principal conclusions. As far as possible, please keep the introduction comprehensible to scientists outside your particular field of research”.
It was also recommend that the introduction be substantially reduced – however, it is unclear if this is done or not in this version.
Can the authors please provide a “clean” version as well as a track changes version to review as I was unable to see if the comments were addressed in the version submitted.
Please re-submit your responses to the questions, ensuring all of the questions and comments are addressed.
Author Response
Reviewer 1
We would like to thank the reviewer for their deliberate reviews of the research article. The raised considerable concerns are very helpful for improving the article. We agree with almost all their comments that we have revised the article and responded to the comments accordingly.
The detailed responses to each of the reviewers’ comments will be stated below. We clearly stated the revised parts with the particular paragraph shown, as well as indicating the page for referring to the paper; or if we have slightly countered with some of the points, we stated the reason with supporting literature. We hope that the reviewers will find our responses persuasive and cogent, and we are willing to accept further suggestions that the reviewers may have.
In the following, each response is targeting each reviewers’ comments, the comments are in italics with responses inserted after it.
This article reported on the redevelopment then the reliability and validity of a Chinese version of the SIT-Q questionnaire for measuring sedentary time. The testing was done in Chinese older adults. Overall, the paper would benefit from considerable editing, both the tighten the argument and remove non-directly relevant information, and also to address the number of comments highlighted below.
- Title: suggest using “Redevelopment and Examination” in the title.
Response: Thank you for your suggestions, changes are done accordingly.
The abstract needs careful editing and proof reading:
- Line 12: seems like there is an additional sentence in here remaining from the template “Place the question addressed in a broad context and highlight the purpose of the study”
- Line 15: clarify where the accelerometer was worn, and whether diary data was collected. Also include the age of the participants.
- Line 17 – the word “conclusions” is here, but the results for study 2 are not presented yet. Report the actual validity results for the accelerometer data with the Sit-Q.
Response: In regard to the suggestions of the reviewer, the required information is added, and the conclusion part is re-written.
- The introduction is currently quite broad, and could be substantially shortened to focus primarily on the evidence gaps in the existing measurement tools, and also why the SIT-Q-7D was chosen. It also needs careful editing, with some of the pick-ups below:
- Lines 23-28 the instruction for the introduction have not been removed.
- Line 82: missing the term adults (…older adults).
- Line 88: clarify what is meant by “particularly in studies with a large sample size”.
- Line 89-91: this sentence is contradictory with both limitations and strengths in the same sentence. Please correct.
- Line 129: Study one examined reliability, study 2 examined validity; however, this line says it is the opposite. Please correct.
Response: In regards to the above suggestions of the reviewer, the required information is corrected. Referring to comment 8, It was referred to the adaptability among large sample studies.
- Introduction: The authors argue in the introduction that different lifestyle patterns may need different cut points for sedentary time. However, it is important to note that there are recommendations to move away from cutpoints, and utilise (for example) machine learning methodologies. It is also somewhat unclear how this paragraph relates to the study report.
Response: Sorry for the confusion, in order to make our point clear, elaboration has been made upon that. The statement regarding the variations of the cut-off points has been bought up because the unstandardized cut-off points for measuring sedentary behavior might have affected the gold-standard reputation of the accelerometer, and therefore, subjective measures’ benefits should be uphold.
- Introduction: please explicitly state what the validity of the questionnaire will be measured against in the final paragraph.
Response: Thank you for the suggestion. The concurrent validity of the questionnaire is measured, and this is stated at the end of the paragraph.
- Methods: Lines 145-159 and Table 1 belong in the results as they are reporting on participant characteristics. Please also use formatting in Table 1 to different the headers from the category (e.g., Gender, Men, Women).
Response: The demographic information was moved to the result session, and the table is reformatted.
- Methods: clarify here whether participants of study were the same as study 1 and if not whether the same recruitment process was used.
Response: The same recruitment process was use in study 2, as stated in line 177.
- Methods, Line 171: seems like an extra line here (they assess sedentary time)
Response: In regards to the suggestions of the reviewer, the required information is corrected.
- Methods: please include sample size calculations.
Response: Taking a meta-analysis outcome (Prince et al., 2020), the correlation between a self-report and device measure of sedentary time as r=0.32, the Gpower software indicated that 74 samples will be required for this reliability and validity analysis.
Prince, S.A., Cardilli, L., Reed, J.L. et al. A comparison of self-reported and device measured sedentary behaviour in adults: a systematic review and meta-analysis. Int J Behav Nutr Phys Act 17, 31 (2020). https://doi.org/10.1186/s12966-020-00938-3
- Methods: Section 2.2.1.2 – it is unclear at this stage why this questionnaire is listed here, as it was not included in the aims of the study. It is also unclear why this was used as a validity measure, and it is unclear when this questionnaire was asked e.g., was it asked after the sit-Q? The order of the questionnaires could impact on the responses. The methods also imply that only one questionnaire was asked. If there was two, then this should be made explicit in the process section.
Response: Regarding the reviewer’s concern, we have changed the sequence of the methodology, by stating the Procedure before the illustration of the included measure. It is important to point out that another questionnaire is involved because, other than the accelerometer, another well-developed Chinese subjective measure should also be involved in the concurrent validity examination as well.
- Methods: section 2.2.3 – make clear whether all participants wore accelerometers or just those in study 2.
Response: This has been made clear in the manuscript, with only participants who were involved in Study 2 wore the accelerometers.
- Methods: please provide reference and data for the validity of hip worn accelerometers for measuring sedentary time, particularly given these are not the gold standard for field-based measurement. Also please provide reference for the non-wear criterion. Please indicate whether diary data was collected to understand any context-specific information.
Response: Based on the reference below (is indicated in the manuscript), the valid wear days are 3 weekdays and 1 weekend, while the non-wear criterion is having a window of 90-min of consecutive vector-magnitude counts per minute. The current study has minimized the 1 weekend due to the older adults’ participants are retired, the differences between workdays and non-workdays might not be significant.
Choi, L., Ward, S. C., Schnelle, J. F., & Buchowski, M. S. (2012). Assessment of wear/nonwear time classification algorithms for triaxial accelerometer. Medicine and science in sports and exercise, 44(10), 2009–2016. https://doi.org/10.1249/MSS.0b013e318258cb36 (Reference number 31)
- Methods: please provide more detail on the elderly centres and who goes there. Are they representative of the broader older adult Chinese population, including the type and time of sedentary activities that would do?
Thank you for your questions. We understand the reviewer’s concern about the representativeness of the sample. It is important to point out that Hong Kong elderly centers are considered as the most effective and representative way of reaching elderly samples in Hong Kong. All older adults could go to these elderly centers. In Hong Kong, they are considered as a kind of community centers, for those who would like to meet friends, join some free or low-cost leisure activities. These centers are mostly funded by the Hong Kong government and NGOs. Despite reaching older adult participants through the elderly centers are considered as more systematic, in the authors’ opinion, the elderly centers might have a possible effect on the sedentary behavior measure of the older adults. It is because, we are not able to access the hidden older adults, who tend not to access any public activities. While this issue has been added in the discussion section.
- Methods: it is unclear why the questionnaire was not asked about the same time period as the monitor wear. This presumes that the behaviour is stable.
Response: We acknowledge the concern of the reviewer. However, from the practical perspective, not every elderly participant was willing to put on the accelerometer for such as a long period of time. Hence, it caused a certain extent of difficulty for us, and so we need to recruit another group of older adults who were willing to put on the device. But still, the participants of Study 2 have also completed the questionnaire, at the same time period the monitor wear.
- Methods: clarify if the accelerometer data was compared to just the totals, or whether it look at correlations with domain-specific measures. Clarify how wear time was taken into account when analysing and reporting the monitor data.
Response: According to the measurement section in the manuscript, it stated that weighted daily mean sedentary time, while a statement is added to the illustration of the SIT-Q-7D that “In the data analysis of concurrent validity, only the total daily sedentary behaviour time was taken into account.”
- Results, line 276-279: this belongs in the methods section.
Response: The lines were moved to the methods section.
- Results, line 279; line 285 the descriptions of Table 2 does not match what is presented in Table 2. The Cronbach’s alpha values and ICCs are not presented. This looks more like Table 5
- Results, Table 2: it is unclear why this is n=38 – isn’t this from study 1?
- Results: as per previous comments, it is unclear why the results for the SBQOA are presented as this was not one of the aims of the study. It is recommended to remove this information and association data or to clarify the role more explicitly in the introduction and methods.
Response: Sorry for the confusion, the table sequence has been re-arranged by changing originally Table 5 into Table 2, with Table 2 specifying the N=84.
- Results: Table 3 shows that accelerometer-derived sedentary time was 3345.58 ± 1126.32 minutes per day. This is not possible – please clarify. Please also add wear time.
Response: The total sedentary behavior time was the total time of the consecutive 7 days. Sorry for the confusion.
- Results, Line 325: as per previous comments, it is unclear why the SBQOA is included, and also then, why it is appropriate to use as a validity measure. These findings do not necessarily indicate true behaviour.
Response: As mentioned earlier, other than the accelerometer, another well-developed Chinese measure was added to study 2, as an indicator of concurrent validity examination. While it might not be a true reflection of behavior, the SBQOA is a well-developed self-report sedentary behavior measure, which could be compared to the SIT-Q-7D, a newly developed Chinese questionnaire.
- Discussion, Line 333: recommend not using the word “controversial”. Just report that there was no significant correlation.
Response: Thank you for the suggestions. The word controversial is deleted.
- Discussion: there is comparison to previous studies in terms of sedentary time; however, it is unclear how wear time was taken into account. Recommend comparing percentage of the day sedentary across the different studies.
Response: We acknowledge the concern of the reviewer. While, sorry for the confusion, the comparison unit stated in the discussion (See Table 7) should be clearly stated, which is the daily average sedentary behavior (in minutes), which means the number of days wearing the devices had been divided.
- Discussion: The questionnaires were interviewer-administered. Are there plans to examine these psychometric properties using self-completion?
Response: Sorry for the confusion, to clarify the issue of interviewer-administrated, in fact, the meaning of interviewer-administrated here was having the research assistants read the item word by word to participants of those who are less educated and less literate. Hence, in order to release this confusion, the procedure part has been edited, and restated to “they completed the questionnaire in a face-to-face context with the assistance of trained researchers (i.e., research assistants).” Therefore, there were no individual differences that might have been caused by different interviewers, and the research assistants were just reading the words according to the item, hence, it was still considered as a self-completion questionnaire.
- Discussion, Line 458-9: this is additional information that should be removed.
Response: Thank you for the suggestions. The lines are deleted.
Second Review Response
The introduction should briefly place the study in a broad context and highlight why it is important. It should define the purpose of the work and its significance. The current state of the research field should be carefully reviewed and key publications cited. Please highlight controversial and diverging hypotheses when necessary. Finally, briefly mention the main aim of the work and highlight the principal conclusions. As far as possible, please keep the introduction comprehensible to scientists outside your particular field of research”.
It was also recommend that the introduction be substantially reduced – however, it is unclear if this is done or not in this version.
Response: The introduction template statement has been removed from the document. Subtitles are added to the Introduction part to clarify and highlight the phenomenon and significance of the current research. We hope the reviewer could find the Introduction more presentable and readable.